# Adherence Level to Arterial Hypertension Treatment: A Cross-Sectional Patient Survey and Retrospective Analysis of the NHS Prescription Database

**DOI:** 10.3390/healthcare9081085

**Published:** 2021-08-23

**Authors:** Anna Gavrilova, Dace Bandere, Konstantīns Logviss, Dins Šmits, Inga Urtāne

**Affiliations:** 1Department of Pharmaceutical Chemistry, Faculty of Pharmacy, Riga Stradins University, LV-1007 Riga, Latvia; Dace.Bandere@rsu.lv (D.B.); Inga.Urtane@rsu.lv (I.U.); 2Red Cross Medical College, Riga Stradins University, LV-1009 Riga, Latvia; 3Baltic Biomaterials Centre of Excellence, LV-1007 Riga, Latvia; 4Department of Dosage Form Technology, Faculty of Pharmacy, Riga Stradins University, LV-1007 Riga, Latvia; konstantins.logviss@gmail.com (K.L.); Dins.Smits@rsu.lv (D.Š.); 5Department of Public Health and Epidemiology, Faculty of Public Health and Welfare, Riga Stradins University, LV-1010 Riga, Latvia

**Keywords:** adherence barriers, blood pressure control, e-health system, electronic prescriptions, intentional non-adherent, Latvia, medicines, Morisky Widget (MMAS-8), patient attitudes, pharmaceutical care

## Abstract

One of the major problems in cardiology practice is poor adherence to antihypertensive medication. This study aimed to evaluate medication adherence; we also aim to investigate the predictors of intentional and unintentional non-adherence. We issued a survey containing questions about patient demographics, blood pressure control, pharmaceutical care, and adherence level to medication. Retrospective analysis of the prescription database of the National Health Service of the Republic of Latvia was performed. The prevalence of non-adherence was 45.9%. The lowest adherence rate (38.2%) was found among patients taking medication for 2–4.9 years. Even though 84.7% of respondents had a blood pressure monitor at home, only 25.3% of them reported measuring blood pressure regularly. There were differences between the groups of adherent patients in terms of the patients’ net income (*p* = 0.004), medication co-payments (*p* = 0.007), and whether the pharmacist offered to reduce the costs of drug therapy (*p* = 0.002). Roughly half of the prescriptions (50.4%) containing perindopril were purchased by patients from pharmacies. The medication adherence level and blood pressure control at home were assessed as low. Intentionally non-adherent respondents discontinued their medication because of fear of getting used to medicines. The pharmacists’ behaviour in terms of offering to reduce the costs of medications used was influenced by socio-economic factors.

## 1. Introduction

High blood pressure (BP) is a serious medical conditions affecting 1.13 billion people worldwide according to the World Health Organization [1]. Untreated arterial hypertension (AH) can increase the risk of heart, brain, kidney, and other diseases (e.g., diabetes mellitus). [2]. AH diagnosis is confirmed when systolic–diastolic BP is ≥140/90. The goal of therapy is to achieve optimal blood pressure, namely a systolic BP of ≤135 mmHg and diastolic BP of ≤85 mmHg when ambulatory [3,4].

According to new concepts in guidelines for the management of AH, wider use of out-of-office BP measurement with ambulatory or home blood pressure monitoring is an option not only to confirm the diagnosis of AH in order to detect “white coat” (when a patient’s blood pressure readings are inaccurate due to certain environments) and masked AH, but also to control the efficacy of the therapy [3]; therefore, it is important that the patient not only has a blood pressure monitor at home, but that they also use it regularly.

Currently, one of the major problems in cardiology practice is poor adherence to drug therapy [2]. Non-adherence is a key reason why the majority of patients do not achieve optimal arterial blood pressure, resulting in an increased risk of cardiovascular complications [2,5]. Age, income, duration of AH, pharmacological treatments, and being a new user of a drug can influence patient adherence rates. Misdiagnoses such as “white coat” hypertension, comorbidity, high medication costs, and side effects may lead to non-adherence [6]. At least 30% of patients aged ≥60 years do not follow the prescribed medication regimen [7,8]. Non-adherent patients can be divided into two subgroups depending on their behaviour—intentional and unintentional. Intentional non-adherence is defined as conscious discontinuing, skipping, or changing of doses of medications, while unintentional non-adherence is associated with accidental forgetting or careless skipping of drug doses or not fully understanding the dosing regimen and patient information leaflet. Intentionally non-adherent patients do not take their medicines due to beliefs and concerns or problems about the medicines, while unintentionally non-adherent patients do not take their medicines because of practical problems [7,9].

There are several ways to improve adherence, including simplification of pharmacological regimens, education and counselling, and development of health programs to help patients [10,11]. The quality of the relationship between the healthcare professional and the patient is an important aspect to encourage the patient to follow the prescribed recommendations [12]. New electronic systems, such as e-health solutions, provide fast and unified electronic communication between health professionals, ensuring collection of a patient’s personal data and basic information about a patient’s health, as well as circulation of electronic prescriptions [10].

The key role of pharmacists in the long-term management of AH through providing education, support, and follow-up of treated hypertensive patients is emphasized as an important part of the overall strategy to improve BP control [3]. Since high BP is a chronic condition, both the drug regimen and other factors affecting it should be monitored continuously, even if patient was diagnosed with AH several years ago. The findings from other studies have shown that pharmacist interventions, such as medication reviews, adherence counselling, and motivational interviews, have positive impacts on the level of adherence [10].

The present study aims to evaluate medication non-adherence and explore its associated variables, such as patient demographics, BP control, pharmaceutical care, and e-health data, among patients with AH in Latvia. The purpose of the study is to investigate dissimilarity between cohorts depending on AH medical treatment duration and intentional or unintentional non-adherence.

## 2. Materials and Methods

### 2.1. Ethics Approval

Ethical approval for the use of data was provided by the Research Ethics Committee of Riga Stradins University. All procedures performed in the study involving human participants were in line with the ethical standards of the institutional and national research committee and with the 1964 Helsinki Declaration and its later amendments or comparable ethical standards. Participation in the study was voluntary and anonymous. The data were collected and processed in accordance with EU Regulation 2016/679 on the protection of natural persons with regards to the processing of personal data.

Prior to being interviewed by staff, participants received verbal and written information about the purpose and structure of the interview in Latvian and Russian and verbally gave their consent to participate in our study. The verbal agreement was acceptable due to respondents’ confidentiality. If a participant agreed to participate in the survey, then they answered the questions. In cases of refusal or unwillingness to continue participating, the respondent was released from the study obligations and the associated data were abolished. The obtained data will be stored in a coded form for 10 years at the Department of Pharmaceutical Chemistry, Faculty of Pharmacy, Riga Stradins University.

### 2.2. Data Collection

#### 2.2.1. Patient Survey

An anonymous patient survey was conducted in several Latvian cities and towns in a primary care setting from 1 September 2018 to 1 April 2019. The survey included patients over 18 years of age taking at least one antihypertensive medication. A total of 187 patients with AH were involved in the study. The respondents’ purpose for visiting the pharmacy was to fill a prescription for antihypertensive medicine. All data were collected via face-to-face interviews after pharmacist consultation at 12 community pharmacies in Latvia. The question selection was based on previously published studies [13,14,15,16]. Interviews were performed and survey forms were filled in by a pharmacist. All pharmacists participating in the data collection were specifically trained. The pharmaceutical care provider and pharmacist performing and completing the survey were different people. Reflecting the nationality demographics, the survey was available in Latvian and Russian, and it took an average of 15–20 min to complete.

#### 2.2.2. Questionnaire

The questionnaire contained 42 questions divided into four sections: (1) patient demographics; (2) AH control; (3) pharmaceutical care; (4) adherence level to medication.

1.The patient demographics included data on age, sex, living status, education, residence, employment status, monthly income after taxes in euros (EUR), body mass index (BMI), smoking status, and physical activity.2.The following items were assessed to characterize the therapy used and to evaluate AH control:How much and how often the patient takes medication on daily basis;Whether the patient takes dietary supplements;Years since the initiation of AH medication use;Whether the patient has a blood pressure monitor at home;Whether the patient’s blood pressure is controlled;Whether there has been a hospitalization related to AH.

One more item was assessed to find out whether the patient had enough information about their condition (AH).

3.Closed-ended questions were selected to characterize pharmaceutical care and to determine whether the pharmacist:Is able to answer all the patient’s questions;Warns the patient about possible side effects and informs them on which medicines should not be taken concomitantly;Asks the patient about the effectiveness of therapy used;Recommends solutions to reduce medication costs;Asks about any disease or medicine used prior to offering any medication or dietary supplement.

An open-ended question was asked to find out how the patient chooses a particular pharmacy to buy their prescribed medications.

4.The participants’ medication adherence level was assessed using Morisky Widget MMAS-8 software. MMAS-8 is a validated survey method with high reliability and validity used to evaluate the level of adherence to medication for chronic conditions such as AH. MMAS-8 consists of eight items, the first seven of which are yes/no questions, while the last one is a five-point Likert scale rating [17,18,19,20,21]:1.Do you sometimes forget to take your hypertension medication?2.People sometimes miss taking their medications for reasons other than forgetting. Thinking over the past two weeks, were there any days when you did not take your hypertension medication?3.Have you ever cut back or stopped taking your hypertension medication without telling your doctor because you felt worse when you took it?4.When you travel or leave home, do you sometimes forget to bring along your hypertension medication?5.Did you take your hypertension medication yesterday?6.When you feel like your symptoms are under control, do you sometimes stop taking your hypertension medication?7.Taking medication every day is a real inconvenience for some people. Do you ever feel hassled about sticking to your hypertension treatment plan?8.How often do you have difficulty remembering to take all your hypertension medications?

Survey answers were coded in Morisky Widget MMAS-8 software. Results were measured by points. In this study, the patients were divided into two groups—adherent and non-adherent—using a six-point “cut point” [16]. After completing the questionnaire, patients with a score of fewer than six points were informed that they were non-adherent. If the value of intentional non-adherence points was higher than non-intentional non-adherence, the patient was classified as intentionally non-adherent, and vice versa. Differences equal to zero were not included in the analysis of non-adherence type detection.

#### 2.2.3. E-Health System

Information on the reimbursed electronic prescriptions for one year (from 1 April 2018 to 31 March 2019) was obtained from the Latvian National Health Service (NHS) database. The prescriptions were selected according to two criteria: the medicine was prescribed within the national reimbursement system for the diagnosis of essential (primary) hypertension (I10) and contained perindopril (as monotherapy or combination therapy). Perindopril was chosen as a selection criterion because according to the statistics on medicines consumption [12], it is one of the most commonly used drugs for the treatment of AH. In addition, multiple medicinal products contain this active substance and are available on the market (monopreparations and fixed-dose combinations) [13].

### 2.3. Data Analysis

Statistical Package for the Social Sciences (IBM SPSS Statistics 23.0, Chicago, IL, USA, 2010; https://www.ibm.com/products/spss-statistics, accessed on 22 August 2021) was used for data analysis. Descriptive statistics were used to measure and calculate frequencies, while percentages were used for categorical variables, means, and standard deviations of the sample. Categorical variables were compared using a Chi-square test. For continuous variables, a *t*-test or Mann–Whitney U test was employed after their normality was determined by Kolmogorov–Smirnov test. One-way ANOVA was used to analyze the differences between the means of more than two groups. All of the statistical tests were two-sided using a significance level of 0.05. Missing data were coded as “missing values” and were not included in the analysis.

## 3. Results

### 3.1. Patient Survey

Out of 187 respondents, 171 completed the full questionnaire. One questionnaire was unsuitable for data analysis because the duration of antihypertensive therapy was not known in one patient. The average age of the 170 respondents was 64.6 years (SD = 13.0). With regards to living status and residence, 118 individuals were sharing their household with someone and 118 were also urban dwellers. Almost half of the respondents (44.7%) had higher education. In total, 46.5% of the respondents were non-working retirees, while the most common net income (50.6%) ranged from 300 to 600 EUR. The vast majority of the patients were non-smokers (84.1%). Obesity was found in 43.1% of the survey participants, whereas more than half of them (61.8%) performed physical activities for less than 150 min a week.

Only 10.6% of the respondents had no concerns about their health, while 38.8% had some doubts about medication use. On average, 3.9 (SD = 2.3) medicinal products and 1.0 (SD = 1.2) dietary supplements were used daily, with 47.1% of patients taking their medicines once daily. Despite the fact that 84.7% of respondents had a blood pressure monitor at home, only 25.3% of them reported measuring blood pressure regularly, while the rest measured it inconsistently or did not measure at all. Here, 74 out of the 170 participants considered that their blood pressure was always controlled. Nevertheless, nearly one-quarter (25.9%) had a health event such as heart failure, stroke, changes in mental status, or chest pain (unstable angina) associated with hypertension that had led to a hospitalization. Most patients (78.9%) were familiar with the recommended diet but only 32.4% of them actually followed it. In total, 75.3% considered that they had been provided with sufficient information on AH, while 55.9% confirmed that their co-payment for the purchased medicines was not high.

Pharmacists explained the use of the medication in 92.4% of cases; however, 9.4% of patients still felt that they did not understand how to use their medicine properly after leaving the pharmacy. In most cases (89.4%) the pharmacist was able to answer all questions asked by the patient and in 65.9% they offered solutions to reduce the costs of drug therapy. The pharmacists rarely (22.4%) wondered whether the current therapy was effective in achieving therapeutic results (to maintain optimal blood pressure). In 37.1% of cases, the pharmacist asked the patient about their disease or medication used prior to offering an over-the-counter medicine or dietary supplement. In total, 65 out of the 170 participants were warned about the possible side effects and informed on which drugs should not be taken concomitantly. The respondents mostly (38.8%) chose a particular pharmacy to buy their medicine because it was located close to their place of residence or work, followed by the quality of pharmaceutical care provided in the pharmacy (27.1%).

In total, 45.9% of the respondents were found to be non-adherent to drug therapy according to the Morisky Widget MMAS-8 software; 36 persons from the non-adherent patient group unknowingly used the drug incorrectly, while 42 persons did it intentionally. Intentionally non-adherent patients were characterized by the following beliefs: they considered they had a health issue (88.0%); they were concerned about the medicines they used (42.9%). Somewhat paradoxically, over one-third of those (13 out of 36) reported concomitant use of two or more food supplements. A detailed description of the cohorts is shown in Table 1.

The proportion of adherent patients decreased to 38.2% with medication use for 2–4.9 years, but the adherence rate increased with the increase in hospitalization episodes due to AH. These trends are demonstrated in Table 2 and Figure 1.

There were differences between the groups of adherent patients in terms of net income (*p* = 0.004), medication co-payment (*p* = 0.007), and whether the pharmacist offered to reduce the costs of drug therapy (*p* = 0.002). The observed findings are shown in Table 3 and graphically in Figure 2. A relationship between adherence to treatment and employment status (*p* < 0.001) was found. Non-working seniors were more adherent (adherence level 64.6%) than other employment groups.

### 3.2. E-Health System

Within one year period, a total of 2.546 million perindopril-containing prescriptions were issued, of which 1.284 million (50.4%) were purchased in pharmacies. Perindopril was mostly (91.7%) prescribed in combination with other active substances (fixed-dose medication) belonging to hypolipidemic agents, calcium channel blockers, or diuretics. Medicines were mostly (96.7%) prescribed by a family doctor (general practitioner), followed by a physician’s assistant (1.1%) and cardiologist (0.7%). Out of all the prescriptions, only in 1577 cases the healthcare professional indicated the international non-proprietary name (INN) in the prescription. It was not possible to determine the exact level of adherence, because prescription information was provided so that the patient could not be identified.

## 4. Discussion

A lack of AH control was disclosed among the respondents. Most patients (84.7%) have a blood pressure monitor at home, but only 25.3% of patients measure their blood pressure regularly. Almost half of the patients without BP measurements stated that they had controlled or normal BP, meaning they might determine optimal blood pressure based on subjective feelings. These subjective criteria cannot give credible evidence that blood pressure is actually controlled and the risk of potential complications is reduced. Three-quarters of the respondents believed they have been given a sufficient amount of information on AH, while 78.9% are aware of the recommended diet. Only one in three patients adhered to a diet, showing that providing the patient with the necessary knowledge is not enough; it is important that they are personally interested in getting information and acting according to the recommendations.

The prevalence of non-adherent patients with AH was 45.9%, which was rated as a high score. The obtained results suggest that medication adherence is still a relevant problem, as almost half of patients do not follow the prescribed drug therapy. The highest proportion of non-adherent patients was observed in the cohort of patients taking AH medication for 2–4.9 years. After >5 years of continuous medication therapy, the patient adherence level tended to increase, which may possibly have been associated with an undesirable medical event (Figure 1), understanding the importance of the medication therapy, or long-term education about AH. Intentionally non-adherent patients discontinued their medication because of fear of getting used to medicines, while at the same time dietary supplements were found to be widely used. This cohort included younger patients of working age, who thought that co-payment for the medication was not high. From these results it can be concluded that co-payment for the medication is not the main obstacle to medication adherence. Patients in this group had doubts about the safety, necessity, or harmlessness of their medication. This might have direct and indirect impacts on patient behaviour. At the same time, they also tended to inform their family doctor less frequently about usage of other medication or food supplements (Table 1). This implies that patients are looking for an alternative to prescribed drugs on their own, resulting in increased self-medication.

The findings related to adherence suggest that adherence should not be treated as a habit, as the proportion of non-adherent patients varies depending on factors such as age, sex, employment status, number of prescribed drugs, attitudes, behaviour, and medication literacy [5,8,22]. Other studies have shown that healthcare professionals, including pharmacists, should identify and address patients’ negative beliefs about medicines to improve adherence rates [23,24]. Pharmaceutical care is of great importance, as it not only provides the patient with medication, but it also ensures pharmacotherapeutic consultation and provides information on medicinal products and their rational use. In the present study, it was found that some respondents chose a particular pharmacy not because of its location, but because of the quality of pharmaceutical care provided there. Nevertheless, more consideration should be given during the consultation to assess the efficacy of the therapy used, existing co-morbidities, drug interactions, and contraindications. It is important for the specialist not only to explain the use of a medicine and its risks, but also to make sure the patient has actually understood the information provided. This is likely to reduce the patient’s doubts about their medications and will possibly increase the level of adherence. We found that pharmacists were more likely to suggest lowering the costs of medicines in patients with a duration of hypertension >10 years. This group predominantly contained non-working seniors with less than €600 in net income. Figure 2 shows that most patients considered the co-payment for their medicines to be high, which is why pharmacists were more likely to offer to reduce the drug costs. The healthcare professionals observed differences in socio-economic status between patient groups. Their compassion and willingness to help patients are significant human factors.

The fact that almost half of the prescriptions were not purchased at the pharmacy raises concerns about potential reasons as to why patients choose not to take the medicine prescribed by their doctor. Patients’ co-payments for brand-name and fixed-dose medications, which were most often prescribed, were higher than for generic medicines. As of 1 April 2020, a regulation was adopted requiring doctors to henceforth indicate the international non-proprietary name within the reimbursement system in Latvia [23]. This means that pharmacists may only provide patients with the reference medicine with the appropriate composition currently available at the pharmacy. Statistics for 2020 show that the consumption of generic medicines has increased, resulting in reduced costs of medicines for patients [24].

The identified problems need to be addressed, and one of the ways to do this is to improve the e-health functionalities. Patient information regarding health status, used medication, and dietary supplements should be readily available to all healthcare professionals involved in the treatment process [25]. It is worth noting that electronic platforms are simply tools for collecting patient data. In order to ensure optimal treatment and achieve better outcomes, an individual approach should be provided for each patient. Productive dialogue is the best way to assess a patient’s condition and clarify their attitude towards a prescribed treatment.

## 5. Study Limitations

Despite analyzing a wide range of sociodemographic e-health data and non-pharmacological interventions, relationships could not be established, since this was a cross-sectional study. Unfortunately, the healthy user bias could not be avoided in this study, as the respondents who agreed to participate in the study were expected to be healthier than non-respondents, i.e., following a healthier diet, being more physically active, avoiding smoking, and taking better care of their health [26]. With respect to this bias, we believe the actual situation could be worse. Additionally, we did not record whether the home BP monitoring devices were validated or which makes of devices were most used.

## 6. Conclusions

The prevalence of non-adherent patients with AH was 45.9. Intentionally non-adherent patients discontinued their medications because of fear of getting used to the medicines. The highest proportion (61.8%) of non-adherent patients was observed in the cohort of patients taking AH medication for 2–4.9 years. Only one in four patients measured their BP regularly, which was generally assessed as a low level of BP control at home. Almost half of the prescriptions were not purchased at the pharmacy, raising concerns about potential reasons as to why patients choose not to take the medicines prescribed by their doctor. Pharmacists as healthcare professionals provide patients with information on the medicinal products, but they are not concerned enough about the factors affecting the therapeutic effect. Socioeconomic factors such as employment status and income level influenced the pharmacists’ behaviour, whereby they offered to reduce the costs of the medications used.

## Figures and Tables

**Figure 1 healthcare-09-01085-f001:**
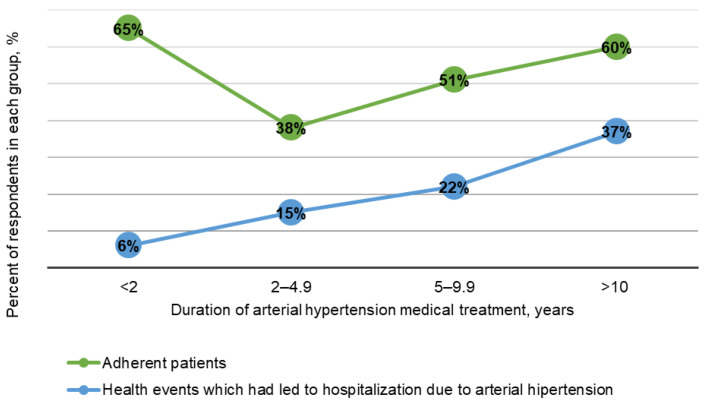
Proportions of adherent patients and health events that led to hospitalization due to AH.

**Figure 2 healthcare-09-01085-f002:**
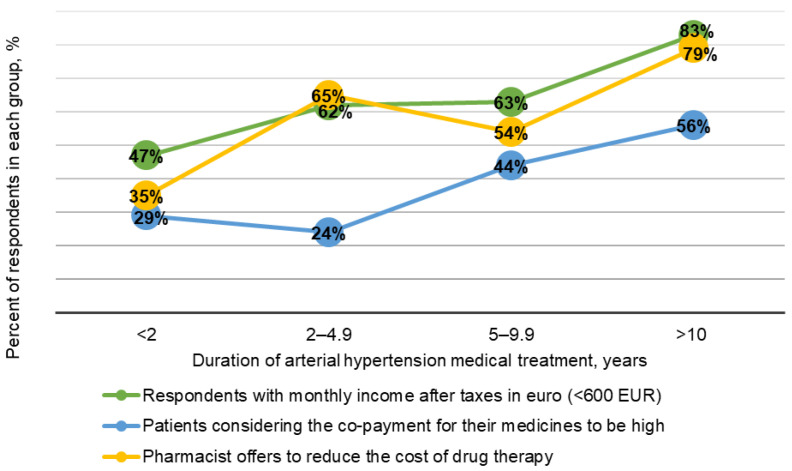
The proportions of adherent patients divided into groups depending on net income, medication co-payment, and whether their pharmacist offered to reduce the costs of drug therapy.

**Table 1 healthcare-09-01085-t001:** Associations between adherence level and different sample characteristics.

Variable	Category	Adherence Level			*p*-Value
		Adherent *	Intentionally Non-Adherent *	Unintentionally Non-Adherent *	
			Years, Mean ± SD		
Age		677 ± 123	586 ± 123	641 ± 131	<0.001
			**Number (n)**		
Sex	Female	73	23	31	0.002
	Male	18	19	5	
	*Odds*	4.06	1.21	6.20	
Health Event that Led to Hospitalization due to AH	Yes	24	6	14	0.048
	No	65	36	22	
	*Odds*	0.37	0.17	0.64	
Patients Considering the Co-Payment for their Medicines to be High	Yes	45	10	20	0.007
	No	47	32	16	
	*Odds*	0.96	0.31	1.25	
Monthly Income after Taxes in Euro	<600	67	24	29	0.032
	≥600	23	18	6	
	*Odds*	2.91	1.33	4.83	
Informs Family Doctor about other Prescribed Drugs from Different Specialists	Yes	57	16	23	0.004
	No	9	12	4	
	*Odds*	6.33	1.33	5.75	
Informs Family Doctor about other Over-the-Counter Drug Use	Yes	28	6	12	0.050
	No	37	27	22	
	*Odds*	0.76	0.22	0.55	
Informs Family Doctor about other Food Supplement Use	Yes	26	6	9	0.165
	No	32	20	15	
	*Odds*	0.81	0.30	0.60	
Has Sufficient Knowledge about AH	Yes	76	29	23	0.059
	No	16	12	13	
	*Odds*	4.75	2.42	1.77	

* Patients were categorized as adherent, intentionally non-adherent, or unintentionally non-adherent based on the MMAS-8 results.

**Table 2 healthcare-09-01085-t002:** Numbers of respondents in each group classified by adherence and health events that led to hospitalization due to AH, depending on the duration of medical treatment.

Variable	Category	Duration of AH Medical Treatment (Years)				*p*-Value
		<2	2–4.9	5–9.9	>10	
		Number (n)				
Health Event that Led to Hospitalization due to AH	Yes	1	5	9	29	0.011
	No	16	28	31	48	
	*Odds*	0.06	0.18	0.29	0.60	
Adherence Level	Adherent	11	13	21	47	0.136
	Non-Adherent	6	21	20	31	
	*Odds*	1.83	0.62	1.05	1.51	

**Table 3 healthcare-09-01085-t003:** The proportions of respondents in each group classified by patient net income, medication co-payment, and whether the pharmacist offered to reduce the costs of drug therapy depending on the duration of AH treatment.

Variable	Category	Duration of AH Medical Treatment (Years)				*p*-Value
		<2	2–4.9	5–9.9	>10	
		Number (n)				
Monthly Income after Taxes in Euro	<600	8	21	26	65	0.004
	≥600	16	28	31	48	
	*Odds*	0.06	0.18	0.29	0.60	
Patients Considering the Co-Payment for their Medicines to be High	Yes	5	8	18	44	0.007
	No	12	26	23	34	
	*Odds*	0.41	0.31	0.78	1.29	
The Pharmacist Offers to Reduce the Cost of Medication Therapy	Yes	6	22	22	62	0.002
	No	11	10	17	16	
	*Odds*	0.55	2.20	1.29	3.88	

## Data Availability

Not applicable.

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
