# Peer review of "Adherence Level to Arterial Hypertension Treatment: A Cross-Sectional Patient Survey and Retrospective Analysis of the NHS Prescription Database"

_healthcare, 2021, doi:10.3390/healthcare9081085_

Round 1

Reviewer 1 Report

The reviewer has no further comments.

Author Response

Thank you for your review.

Reviewer 2 Report

It seems the manuscript revised a lot compared with the last time I received. The revised manuscript is more scientific soundness now, although my questions were not totally answered , but I think is already suitable for publication.

Author Response

Thank you for your review.

Reviewer 3 Report

Specific comments:

  1. In the abstract, the authors wrote (lines 18-19): “There was a trend for the proportion of adherent patients to decrease to 38.2% with medication use for 2–4.9 years.” The context of this result is not clear. What trend is referred to in this sentence? This was a cross-sectional analysis, so it should be clarified what trend refers to and from what the adherence decreased. Where does this 2-4.9 years fit in? The authors should rephrase this sentence.
  2. The authors state in the first sentence of the Introduction (line 33) that high blood pressure is a chronic disease (also line 53 in introduction). This is debatable because hypertension is the chronic condition. Is it a disease though? Or perhaps only a risk factor for CVD? The statement by the authors is contradicted in the next sentence of the introduction (line 35) and also in line 37 where they indicated that AH is a preventable risk factor.
  3. The introduction is too long and deviates from the main aim of the study, especially the paragraph in lines 76-85.
  4. In the methods, have the authors also recorded whether the home BP monitoring devices are validated and which brand (e.g., Omron) devices were mostly used?
  5. In the results section (line 212), what does “equal number of people (69.4%)” mean? How is 69.4% an equal number of people?
  6. The authors can include some of the events associated with hypertension referred to in line 226.
  7. The authors used the word “gender” in their study. How was this defined and determined in this study? Gender refers to self-identification. The authors should consider to change this to “sex”.
  8. The authors should refrain from using the words “statistically significant” and including the p-value. It is redundant to write both. Better to only provide the p-value instead.
  9. In the discussion (lines 313 – 314) the authors mention that only one in three patients adhered to diet. This is not specific. Diet was not assessed? What type of diet and how was this analysed?
  10. The paper ends with the study limitations and not a conclusion. This is strange and should be corrected.

Technical comments:

  1. Please revise the sentence in lines 16-17 of the abstract, it is incomplete.
  2. Please revise the sentence in lines 37-38 of the Introduction, it is incomplete.
  3. Inconsistent use of the abbreviation AH. Sometimes written out (see line 38) and sometime abbreviated (see line 36). Correct everywhere in the manuscript.
  4. Please revise the grammar in lines 46-47 of the Introduction.
  5. Check punctuation in the paper (see e.g., line 73).

Author Response

Specific comments:

1. In the abstract, the authors wrote (lines 18-19): “There was a trend for the proportion of adherent patients to decrease to 38.2% with medication use for 2–4.9 years.” The context of this result is not clear. What trend is referred to in this sentence? This was a cross-sectional analysis, so it should be clarified what trend refers to and from what the adherencedecreased. Where does this 2-4.9 years fit in? The authors should rephrase this sentence.

We accept your comment and have rephrased the sentence to the following text: The lowest adherence rate (38.2%) was found among patients taking medication for 2–4.9 years.

2. The authors state in the first sentence of the Introduction (line 33) that high blood pressure is a chronic disease (also line 53 in introduction). This is debatable because hypertension is the chronic condition. Is it a disease though? Or perhaps only a risk factor for CVD? The statement by the authors is contradicted in the next sentence of the introduction (line 35) and also in line 37 where they indicated that AH is a preventable risk factor.

Yes, we agree with the reviewer’s comment that blood pressure is a condition in which the blood vessels have persistently raised pressure. We have changed the text and it now reads: High blood pressure (BP) is a serious medical condition.

3. The introduction is too long and deviates from the main aim of the study, especially the paragraph in lines 76-85.

We agree with the reviewer’s comment and the introduction has been reduced as advised not to distract the reader from the main objectives of the study. Changes have been made in the manuscript.

4. In the methods, have the authors also recorded whether the home BP monitoring devices are validated and which brand (e.g., Omron) devices were mostly used?

Thank you for your comment. In this study, we did not account what type and brand of blood pressure monitoring device the respondent has. It was important for us to ensure the presence of the device and the regularity of its use. We agree with the reviewer that it might be of importance and have included the following sentence in the Study limitations section: Also, we did not record whether the home BP monitoring devices were validated, and which makes of devices were mostly used.

5. In the results section (line 212), what does “equal number of people (69.4%)” mean? How is 69.4% an equal number of people?

We agree with the reviewer’s comment and have the text. Text “equal number of people (69.4%)” has been deleted and now it reads: With regards to living status and residence, 118 persons were sharing their household with someone, and 118 number of people were urban dwellers. 

6. The authors can include some of the events associated with hypertension referred to in line 226.

We agree with the reviewer’s comment and now the sentence reads as follows (added text is underlined): Nevertheless, nearly a quarter (25.9%) had a health event such as heart failure, stroke, changes in mental status, or chest pain (unstable angina), associated with hypertension that had led to a hospitalization.

7. The authors used the word “gender” in their study. How was this defined and determined in this study? Gender refers to self-identification. The authors should consider to change this to “sex”.

We agree with the comment and changes have been made in the manuscript.

8. The authors should refrain from using the words “statistically significant” and including the p-value. It is redundant to write both. Better to only provide the p-value instead.

We agree with the comment and changes have been made in the manuscript.

9. In the discussion (lines 313 – 314) the authors mention that only one in three patients adhered to diet. This is not specific. Diet was not assessed? What type of diet and how was this analysed?

A specific diet was not mentioned in the questionnaire. This was a generalized question on any kind of diet changes – does the patient reduce salt, alcohol, red meat products use and increase consumption of vegetables, fresh fruits, whole grains, soluble fiber, fish, nuts, and olive oil in his/her nutrition.

10. The paper ends with the study limitations and not a conclusion. This is strange and should be corrected.

We agree with the comment and the conclusion part is added to the manuscript.

Technical comments:

  1. Please revise the sentence in lines 16-17 of the abstract, it is incomplete.
  2. Please revise the sentence in lines 37-38 of the Introduction, it is incomplete.
  3. Inconsistent use of the abbreviation AH. Sometimes written out (see line 38) and sometime abbreviated (see line 36). Correct everywhere in the manuscript.
  4. Please revise the grammar in lines 46-47 of the Introduction.
  5. Check punctuation in the paper (see e.g., line 73).

All technical comments have been revised in the manuscript.

Round 2

Reviewer 3 Report

All comments were appropriately addressed.

This manuscript is a resubmission of an earlier submission. The following is a list of the peer review reports and author responses from that submission.

Round 1

Reviewer 1 Report

The current manuscript entitled "Adherence Level to Arterial Hypertension Treatment: a Cross-Sectional Patient Survey and Retrospective Analysis of the NHS Prescription Database" reports a study done by Anna et al. The authors clearly describes their study objectives, selection criteria and various considerations to identify and check for the factors influencing the possible non-adherence patterns among the AH-drug users. Non-adherence is one of the most important influence factors that needs to be addressed seriously, as it determines the course and effectiveness of any given diseases or codition. The authors selected a one year window pattern for their current prescription and reimbursement patterns. As with most of the studies the pharmacists educating the patients directly influence the outcome of drug usage and stress the importance of them in the health care system.

As many studies proved and even authors have noted the importance of non-pharmacological interventions, it will interesting on the authors part to include some of the common, easy to follow non-pharmacological approaches for the patients to follow and which can influence the disease and drug adherence pattern.

The reviewer doesn't have any major comments/concerns with the study.

Reviewer 2 Report

  1. Title: Adherence level to arterial hypertension treatment: a cross-sectional patient survey and retrospective analysis of the NHS prescription database

  1. General comments
  • This study demonstrated adherence level of arterial hypertension treatment using the database of the National Health Service of the Republic of Latvia. Non-adherence of hypertensive medication was high and home blood pressure control levels were inadequate.
  • This study is interesting to me but I am not sure that methodologic approach is appropriate to prove whether arterial hypertension medication adherence is low or high.
  1. Detailed comments
  • Keywords: Please use MeSH terms in alphabetical order
  • Introduction: Line 37-39: Optimal target blood pressure is different among guidelines and according to age group and patients with other diseases such as chronic kidney disease and diabetes.
  • I cannot understand why patients should measure their blood pressure everyday.
  • Figure 2 and 3: To investigate the difference according to time, you have to perform Chi square analysis and insert p-value. In addition, trend analysis and p for trends should be conducted.

Reviewer 3 Report

The authors wanted to investigate the adherence rate of the patients prescribed with antihypertensive medication.However there are some major points raised my concerns:

  1. Authors need to describe the " white coat hypertension" which usually misdiagnosed as real essential hypertension leading to the side effect of low blood pressure after taking antihypertensive medication in this case, which is the major cause of poor adherence rate. Other side effect must also be described.
  2. There are missing information of the blood pressure level, which is the main reason of prescribing anti hypertensive medication, and also the major information to convince patients to take anti hypertensive medication.
  3. Comorbidity is also an important information. Authors special mention about the ACEI perindopril, which has an indication on post myacardial infarction  heart failure, or diabetic nephropathy.